# Comparative Study of Commercial Silica and Sol-Gel-Derived Porous Silica from Cornhusk for Low-Temperature Catalytic Methane Combustion

**DOI:** 10.3390/nano13091450

**Published:** 2023-04-24

**Authors:** Clement Owusu Prempeh, Ingo Hartmann, Steffi Formann, Manfred Eiden, Katja Neubauer, Hanan Atia, Alexander Wotzka, Sebastian Wohlrab, Michael Nelles

**Affiliations:** 1Department of Thermochemical Conversion, DBFZ—Deutsches Biomasseforschungszentrum Gemeinnützige GmbH, Torgauer Straße 116, 04347 Leipzig, Germany; steffi.formann@dbfz.de (S.F.); manfred.eiden@dbfz.de (M.E.); michael.nelles@dbfz.de (M.N.); 2Department of Agriculture and Environmental Science, University of Rostock, Justus-von-Liebig-Weg 6, 18059 Rostock, Germany; 3Leibniz-Institute for Catalysis e.V. (LIKAT), Albert-Einstein-Str. 29a, 18059 Rostock, Germany; katja.neubauer@catalysis.de (K.N.); hanan.atia@catalysis.de (H.A.); alexander.wotzka@catalysis.de (A.W.); sebastian.wohlrab@catalysis.de (S.W.)

**Keywords:** sol-gel, cornhusk, support material, biogenic silica, low-temperature catalytic methane combustion, Pd/CeO_2_, characterization

## Abstract

The synthesis and characterization of sol-gel-derived cornhusk support for low-temperature catalytic methane combustion (LTCMC) were investigated in this study. The prepared cornhusk support was impregnated with palladium and cerium oxide (Pd/CeO_2_) via the classical incipient wetness method. The resulting catalyst was characterized using various techniques, including X-ray diffraction (XRD), N_2_ physisorption (BET), transmission electron microscopy (TEM), and hydrogen temperature-programmed reduction (H_2_-TPR). The catalytic performance of the Pd/CeO_2_/CHSiO_2_ catalyst was evaluated for methane combustion in the temperature range of 150–600 °C using a temperature-controlled catalytic flow reactor, and its performance was compared with a commercial catalyst. The results showed that the Pd/CeO_2_ dispersed on SiO_2_ from the cornhusk ash support (Pd/CeO_2_/CHSiO_2_) catalyst exhibited excellent catalytic activity for methane combustion, with a conversion of 50% at 394 °C compared with 593 °C for the commercial silica catalyst (Pd/CeO_2_/commercial). Moreover, the Pd/CeO_2_/CHSiO_2_ catalyst displayed better catalytic stability after 10 h on stream, with a 7% marginal loss in catalytic activity compared with 11% recorded for the Pd/CeO_2_/commercial catalyst. The N_2_ physisorption and H_2_-TPR results indicated that the cornhusk SiO_2_ support possessed a higher surface area and strong reducibility than the synthesized commercial catalyst, contributing to the enhanced catalytic activity of the Pd/CeO_2_/SiO_2_ catalyst. Overall, the SiO_2_ generated from cornhusk ash exhibited promising potential as a low-cost and environmentally friendly support for LTCMC catalysts.

## 1. Introduction

Energy generation from fossils deviates from the culture of sustainability and environmental protection due to the wide emissions of greenhouse gases such as methane (CH_4_), nitrous oxide (N_2_O), and carbon dioxide (CO_2_) [1,2]. With the daily increase in global dependency on fossils for energy generation due to industrialization, the need for alternative energy sources has become paramount. Although methane has a relatively short life span in the atmosphere, it possesses a global warming potential 28–34 times higher than that of CO_2_ [3,4], and its abatement is still not realized in the industrial domain due to the lack of feasible and economical mitigation routes [5]. Consequently, the EU [6] and African Unions [7] have enacted stringent policies that serve as a driving force toward the incorporation of carbon-neutral-based resources such as biomass in energy and advanced materials generation. These measures are expected to serve as a framework for reducing the global emissions of greenhouse gases from various industrial and human activities by 2050 [8]. One such intervention is the conversion of methane to a less potent gas, such as CO_2_, via the so-called low-temperature catalytic methane combustion (LTCMC) process [9,10]. 

LTCMC technology activates methane at low temperatures, forming complex intermediate species that are oxidized to produce energy, CO_2_, and H_2_O [11]. Compared with traditional high-temperature combustion, LTCMC operates at lower temperatures, reducing the formation of harmful pollutants such as nitrogen oxides (NOx) and particulate matter [12]. An example of the LTCMC process is the three-way catalytic converter in natural gas vehicles [13]. While three-way catalytic converters reduce CO, HC, and NOx emissions, they are less effective at reducing CH_4_ emissions [14]. CH_4_ is a non-polar molecule and the least kinetically reactive molecule, with a C–H bond strength of 104 kcal/mol [15,16]. This necessitates a high activation energy and decomposition temperature for the homogeneous combustion of methane at approximately 1200 °C [17,18]. Moreover, a typical three-way catalytic converter operates in a nominal temperature range of 100–600 °C [9], which may not be sufficient to achieve the level of methane conversion necessary to significantly reduce emissions. Thus, significant research has been aimed at exploring novel catalyst materials, such as metal-organic frameworks (MOFs), which have tuneable chemical properties to activate and mitigate CH_4_ emissions at low temperatures [19,20]. These materials, including transition metals, oxides, sulfides, carbides, and zeolites, are better suited to promoting methane oxidation at lower temperatures, leading to improved conversion rates and lower vehicle emissions [21,22,23].

In this regard, catalysts with exceptional activity, such as noble metals Pd and Pt, have been reported as the best candidates to promote LTCMC operations [24,25,26]. Nonetheless, the efficiency of these metal oxides (MOx) can be fine-tuned by supporting them on alumina (Al_2_O_3_) [27], sol-gel-based porous silica (SiO_2_) [28], ZrO_2_ [16], and SnO_2_ [29]. These supports act as carriers and play a crucial role in the activity of the metal oxides by providing a high surface area, which enhances the metal oxide dispersion, thermal and mechanical stability, and lifetime of the catalyst [13]. According to Eguchi et al. [30], the performance of a Pd-containing catalyst is related to the nature of the support through Pd dispersion and Pd-support interaction. 

Pd supported on alumina (Pd/Al_2_O_3_) is widely perceived as the ‘standard’ catalyst for LTCMC operations, with various studies providing historical insight into the role and performance in methane catalytic reactions [27,31]. However, Al_2_O_3_ has been reported to be less efficient due to the inert nature of the support and only being active at medium temperatures above 400 °C [30]. More importantly, it often suffers from severe deactivation in the presence of water vapor (10–15 vol%) during operations in the exhausts due to the hydroxylation of the support [32]. The hydroxylation of the support causes the active sites of the Pd catalyst, PdO_x,_ to sinter, forming the less active sites, Pd(OH)_2_. This slows the exchange of active lattice oxygen replacement, resulting in a loss of the catalytic activity [33]. This process is often mitigated by the introduction of Pt, which forms a bimetallic phase of Pd-Pt to slow down the rate of PdO sintering [33]. Although the Pd–Pt bimetallic phase enhances the stability of the catalyst system against sintering and hydroxylation by water vapor, the catalytic activity is usually compromised, as the Pd–Pt phase is less active compared with PdO_x_ under low temperature and oxygen-rich conditions [34,35]. To ensure a high catalytic activity with favorable interactions between the active species and the support, precursors such as ceria (CeO_2_) are incorporated as a promoter into the Pd lattice to form Pd/CeO_2_ complex [36]. The CeO_2_ provides abundant active surface oxygen species, improving the reducibility and thermal stability of the catalyst. However, the effects of high temperature on the catalytic activity cannot be ruled out. According to Peng et al. [37], the active PdO_x_ species decompose into the less active nanometric Pd^0^ species at high temperatures, resulting in a loss of catalytic efficiency. 

For the preparation of novel methane combustion catalysts, the adoption of controlled porous glass (CPG) as a support for the dispersion of the active species was employed [38]. However, the synthesis process of CPG from phase-separated borosilicate glass is expensive and time-consuming, limiting broad industrial applicability [39]. Similarly, the functionalization of Pd on commercial silica supports, such as fused and precipitated silica, synthesized via the hydrolysis process of tetraethyl orthosilicate, provides a feasible alternative to obtain efficient Pd/SiO_2_-based catalysts, albeit with energy-intensive and environmentally unfriendly properties [40]. These downsides render commercial silica less desirable from cost and environmental considerations as a catalytic support material, and therefore necessitates research into other sustainable silica sources that may have the potential to offset these deficiencies. Recent research has shown that using biogenic silica as a catalyst support can lead to higher catalytic activity and selectivity in the LTCMC process [39]. Biogenic silica is a low-cost, environmentally friendly alternative to industrial-produced silica and has been shown to improve the performance of catalysts in LTCMC applications. Liu et al. [39] investigated rice husk-derived porous silica to support Pd and CeO_2_ for LCTCM operations. Although they reported a high catalytic activity for the gas stream at dry conditions, the catalyst had lower activity when tested in the wet feed gas (10.5 vol%-H_2_O). 

Cornhusk, as an agricultural residue, is therefore ideal for use in a holistic manner, not only thermally but also materially for the production of porous biogenic silica [41]. As a result, Prempeh et al. [42] synthesized high-quality biogenic silica nanoparticles using the sol-gel polymeric route with high potential in catalysis operations. Sol-gel products benefit from favorable properties such as high surface area with a bimodal pore size distribution, allowing for adequate surface for the active species immobilization and unimpeded mass transfer of gaseous species (reactants and products) [43]. In addition, they exhibited high thermal stability and hydrophobicity that could reduce hydroxylation and water deactivation during catalysis operations [44]. According to Schwarz et al. [45], sol-gel products have extremely low thermal conductivity, good texture, and excellent stability at high temperatures. Notwithstanding, studies on the incorporation of sol-gel-derived biogenic silica from agricultural residues into the field of catalysis are limited. To the best of the authors’ knowledge, no studies have been conducted to that effect, and this present study serves as a baseline to bridge this knowledge gap.

Against this background, investigations were conducted into the catalytic activity of sol-gel-derived porous biogenic silica from cornhusk as a support for the MOx couple (Pd/CeO_2_) for LTCMC. The catalytic activities of Pd and CeO_2_ impregnated on the sol-gel-derived porous biogenic silica were discussed and compared with conventional support (CWK Köstropur^®^ 021012). Both supports were impregnated with the catalytic materials using the classical incipient wetness impregnation method. The final synthesized catalysts were both characterized, and their catalytic performances of methane combustion were examined under lean methane conditions (800 ppm) in a simulated real and dry exhaust gas in the temperature range of 150–600 °C. Catalytic stability tests of the catalysts were performed by running the reaction at 500 °C for 10 h on stream. This present investigation could offer a sustainable and cost-effective approach to catalyst synthesis, while also contributing to waste reduction.

## 2. Materials and Methods

### 2.1. Materials

Two kinds of supports were used in this study: biogenic silica and commercial silica. Cornhusk (*Zea mays*) residues were obtained from a local farm in the Ashanti region, Ghana, in the form of five round bales, each approximately 50 kg, for a total of 250 kg. Upon collection, they were washed in an extractor (STAHL ATOLL 290 E, Gottlob Stahl Wäschereimaschinen GmbH, Sindelfingen, Germany) at 50 °C for 2 h to remove dirt and soil particles. Commercially available support (99.89 wt.% SiO_2_, CWK Köstropur^®^ 021012, Chemiewerk Bad Köstritz GmbH, Bad Köstritz, Germany,) was also purchased and used as a benchmarked catalyst. Metal oxide precursors of Pd(NO_3_)_2_·2H_2_O and Ce(NO_3_)_3_·6H_2_O were purchased from Sigma-Aldrich (Taufkirchen, Germany) to synthesize the Pd/CeO_2_ nanoparticles.

### 2.2. Preparation of Sol-Gel-Derived Cornhusk Support

The cornhusk residues were combusted at 600 °C for 2 h to obtain unmodified ash (53 wt.% silica content, surface area = 88 m^2^/g, and pore volume = 0.25 cm^3^/g). The unmodified ash was subjected to a sol-gel polymeric route, as detailed in our earlier publication [43], with slight modification to the washing step. A dissolution process of the unmodified ash in NaOH solution at 100 °C for 1 h yielded a sodium silicate solution, which, after a careful pH-controlled process and vigorous washing steps, promoted the formation of silica gels. The gels were dried at 80 °C for 24 h to obtain a silica xerogel powder with improved properties (99 wt.% silica content, surface area = 384 m^2^/g, and pore volume = 0.35 cm^3^/g) compared with those of the unmodified ash. The obtained cornhusk support was characterized and subsequently used as the support for the impregnation of the MOx in the subsequent catalysis experiments.

### 2.3. Preparation of Supported Catalysts

Supported catalysts were prepared via incipient wetness impregnation as reported in the study by Liu et al. [39]. Molten Ce(NO_3_)_3_·6H_2_O was first impregnated into the supports (cornhusk and commercial silica) at 235 °C and dried at 90 °C overnight. The impregnated supports containing CeO_2_ were subsequently calcined at 450 °C at a heating rate of 5 °C/min for 2 h. Hereafter, the resultant CeO_2_/SiO_2_ mixture was impregnated with an aqueous solution of Pd(NO_3_)_2_·2H_2_O. The Pd loading on the supports was controlled to yield 1 wt.% of Pd in the final catalysts. Finally, the catalyst precursors were dried at 120 °C for 12 h and calcined at 500 °C for 1 h. Synthesized catalysts were designated as Pd/CeO_2_/CHSiO_2_ and Pd/CeO_2_/commercial to represent the catalysts impregnated on cornhusk and commercial silica, respectively. 

### 2.4. Characterization Techniques 

The elemental composition of the supports, actual Pd and Ce loadings on the synthesized catalysts (Pd/CeO_2_/CHSiO_2_ and Pd/CeO_2_/commercial) were measured using inductively coupled plasma-optical emission spectrometry (ICP-OES, CETAC, ASX-520, Omaha, NE, USA).

SEM measurements were performed to observe the morphology of the synthesized catalysts. Samples were analyzed by a field emission scanning electron microscope (SEM, MERLIN^®^ VP Compact, Co. Zeiss, Oberkochen, Germany) equipped with an energy dispersive X-ray (EDX) detector (XFlash 6/30, Co. Bruker, Berlin, Germany). Representative areas of the samples were analyzed and mapped for elemental distribution on the basis of EDX-spectra data by QUANTAX ESPRIT Microanalysis software (version 2.0). Samples were mounted on a heavy metal-free Al-SEM-carrier (co. PLANO, Wetzlar, Germany) with adhesive conductive carbon tape (Spectro Tabs, TED PELLA INC, Redding, CA, USA) and coated with carbon (5.0 nm thickness) under vacuum (CCU 010 HV-Coating Unit, Co. Safematic GmbH, Zizers, Switzerland). SEM images were taken from the selected regions with the conditions of an applied detector, accelerating voltage, and working distance indicated on the SEM micrographs.

An FTIR spectrometer (PerkinElmer, Solingen, Germany) was used to identify the types of functional groups present in the synthesized catalysts. The spectrum scope in the range of 400–4000 cm^−1^ with a resolution factor of 1 cm^−1^ was recorded after four scans and background subtraction.

The specific surface area and pore size distribution were determined using nitrogen adsorption/desorption measurements (BET method) in the autosorb iQ-MP/XR apparatus, Quantachrome, Boynton Beach, FL, USA. According to the literature, the catalysts were first degassed for 12 h at 250 °C under a vacuum to remove adsorbed water molecules on the surface and within the pores [46]. The specific surface area was determined by multipoint Brunauer–Emmett–Teller (BET) surface area analysis in the pressure range of p/p_0_ = 0.05–0.30 at 77 K and considering the cross-section area of N_2_ molecules of 16.2 Å [47]. The pore volume and pore size distribution were determined using the nonlocal density functional theory (NDLFT) method and considering the adsorption and desorption branch of the isotherm data. 

X-ray diffraction (XRD) was used to determine the crystalline phases and crystal structure of the Pd/CeO_2_ nanoparticles and the supported catalysts using an X-ray powder diffraction apparatus (XRD, Malvern Panalytical GmbH, Kassel, Germany) equipped with Ni-filtered, Cu-Kα radiation (λ = 1.54 Å). XRD powder patterns were recorded on a Panalytical’X’Pert θ/2θ-diffractometer equipped with an Xcelerator detector using automatic divergence slits and Cu kα_1_/α_2_ radiation (40 kV, 40 mA; λ = 0.15406 nm, 0.154443 nm). Cu beta-radiation was excluded using a nickel filter foil. The measurements were performed with 0.021 s^−1^ and 0.005 s^−1^, respectively. Samples were mounted on silicon zero background holders. Obtained intensities were converted from automatic to fixed divergence slits (0.25) for further analysis. The size of the coherent scattering region (CSR) was determined using the Scherrer equation [48] applied to corresponding phase reflections. The diffraction patterns were collected in the 2θ range from 5° to 80°.

The redox properties of the catalysts were measured by employing hydrogen temperature programmed reduction (H_2_-TPR) experiments with an AC 2920 equipped with a CryColler-unit (Mircomertics, Waltham, MA, USA). The experiments were performed after the following protocol: 80 up to 100 mg of each sample was preheated to 500 °C for 30 min in synthetic air (50 mL/min, 20 °C/min). The measurement was started after cooling down to −20 °C. Each sample was heated up to 800 °C in a mixture of 5% H_2_ in Ar (20 mL/min, 5 °C/min), and the reduction was performed for 30 min. An online thermal conductivity detector was used to measure the hydrogen consumption throughout the entire experiment. The amount of H_2_ consumption was calculated after calibration of the thermal conductivity detector (TCD).

### 2.5. Catalytic Activity Tests for Methane Combustion 

The catalytic performance of the synthesized catalysts was tested in a temperature-controlled catalytic flow reactor (stainless steel V4A, Ø12 mm). A total of 0.2 g of the synthesized catalysts was mixed with 1.2 g of corundum (inert material) and packed into the reactor tube supported on a quartz wool bed. The amount of inert material mixed with the catalyst was chosen to fill a 7 cm length in the catalyst bed. Light-off tests were carried out by heating the catalyst stepwise from RT to 600 °C (10 °C /min heating ramp) under 70 mL/min flow of a simulated synthetic flue gas mixture containing 800–1000 ppm CH_4_, 1528 ppm CO, 207 ppm NO, 10 vol.% CO_2_, 6 vol.% O_2_ balanced with N_2_ (dry condition), or a mixture of above compositions + 12 vol.% H_2_O (wet condition). For each activity run conducted, a gas hourly space velocity (GSHV) of 87,000 mLg^−1^ h^−1^ was used. The conversion of methane ($X_{CH_{4}})$ was calculated as:(1)$$X_{CH_{4}}\left(\%\right)=\frac{CH_{4,\;t=0}-CH_{4,\;t}}{CH_{4,\;t=0}}\times 100$$

The catalytic activities of the catalysts were evaluated based on the temperature at which 50% conversion of methane was achieved (T_50%_). Kinetic studies were conducted with methane conversions values ≤10% to exclude the effects of mass and heat transfer limitations. The stability tests or time-on-stream of the catalysts were performed by maintaining the samples at 500 °C for 10 h on stream.

## 3. Results and Discussion

### 3.1. Structure and Properties of Synthesized Catalysts

Table 1 presents the ICP-OES results of the bulk elemental compositions of the prepared silica xerogel (SX) support obtained from the cornhusk ash and commercial silica. The cornhusk support was characterized by a high silica content (>99 wt.%) with negligible impurities, as later verified in the EDX spectra of the final catalysts and shown in Figure 1. The amount of Pd and Ce loadings on both supports, as measured by ICP-OES, are shown in Table 1. Both synthesized catalysts showed approximately equivalent amounts of the designated concentrations of impregnated Pd, highlighting the effectiveness of the synthesis method in preparing the catalysts. 

The spatial distributions of the Pd and Ce nanoparticles on the surfaces of the prepared catalysts were observed using different selected areas in the SEM/EDX diagrams, and their corresponding peaks of the various elements in EDX are shown in Figure 1. Additional information on the SEM/EDX mappings and spectra values of Pd/CeO_2_/CHSiO_2_ and Pd/CeO_2_/commercial catalysts in atomic wt.% are provided in the Appendix A. As observed in the SEM/EDX mapping images in Figure 1, the Pd grains were non-uniformly distributed in both catalysts as dispersed crystallites over the CeO_2_ granules (red regions on the blue and white surfaces), indicating a successful impregnation of the Pd and Ce species on the supports. Furthermore, apparent proximity between the Pd and Ce can be observed according to the backscattered electrons detector (BSE) and line scan analyses provided in Appendix A, indicating a successful preparation of the decoupled Pd species from Ce on the samples. Additional elements, such as C (from the sample holder) and O, can also be detected in the EDX spectra diagram in Figure 1.

The surface morphologies of the prepared catalysts are examined in Figure 2 by comparing their SEM micrographs, and the results revealed different morphological structures of both catalysts. The Pd/CeO_2_/CHSiO_2_ catalyst exhibited non-uniform shapes and particle sizes (Figure 2a) compared with those of the Pd/CeO_2_/commercial, which was more analogous in its particle sizes and shapes (Figure 2b). In addition, the Pd/CeO_2_/CHSiO_2_ had rough and irregular surfaces with visible pores and cracks, while the commercial silica support exhibited a smooth and uniform surface with less conspicuous pores. The rough surfaces of the cornhusk support could serve as a scaffold and anchor for the impregnation of the active species [39]. It can also be observed from the SEM image in Figure 2c that the Pd/CeO_2_/CHSiO_2_ catalyst exhibited a compact assemblage with narrow pore cavities compared with Pd/CeO_2_/commercial in Figure 2d, which could serve as a constrainer for growing nanoparticles inside the pores [49].

The FTIR spectra of corn husk support (silica xerogel), Pd/CeO_2_/CHSiO_2_, and Pd/CeO_2_/commercial catalysts are shown in Figure 3. The broad peak between 3500 and 3000 cm^−1^ of the cornhusk support is attributed to the asymmetric stretching of O–H groups from the silanol group [43]. This is formed due to water molecules’ adsorption on the silica xerogel’s surface and the occlusion of intermediate species [(OR)_3_–Si–(OH)] within the porous structure during the hydrolysis process of the sol-gel polymeric route [50]. However, this band disappears with the thermal treatment of the catalyst at 500 °C, as seen in the spectrum of the Pd/CeO_2_/CHSiO_2_ after synthesis. Lopez et al. [51] reported similar observations during their study on preparing high surface area sol-gel Pd/SiO_2_ catalysts. Similarly, the infrared spectra between 1644 and 1573 cm^−1^ of the cornhusk support correspond to Si–H_2_O flexion and the bending of the H–O–H [52].

The highest energy bands spectrum centered at approximately 1090–1000 cm^−1^ are assigned to the asymmetric vibration of Si–O–Si bonds [53]. According to Brinker et al. [54], the intensity and symmetry of the high-energy bands could change due to the formation of siloxane bridges. Simultaneously, other characteristic peaks were also visible at 900–500 cm^−1^, which are features of the asymmetric vibration of O–Si–O formed due to the condensation reaction between neighboring silanol groups [43,53]. These silanol groups are significant for the structural characterization of solids, as they are involved in changes in the microcrystallinity of the solids after the post-gelation period [51]. The lower energy bands within the 800 cm^−1^ result from the symmetric vibration of –Si–OH [55,56]. More importantly, calcination at/above 450 °C often leads to the formation of small-energy bands cantered at 485–500 cm^−1^, which are usually assigned to a Pd–O bond [57], as observed in Figure 3. This indicates the successful formation of metal–oxygen bonds through interactions with the support and the presence of Pd active species on the surface of the catalyst. 

The textural properties of the supports and prepared catalysts were measured using nitrogen physisorption experiments, and the results are shown in Figure 4a,b and summarized in Table 2. Both supports exhibited type IV(a) hysteresis, primarily observed in mesoporous materials [49]. However, the shapes of the hysteresis loops were different, indicating different pore structures in the catalysts. For the commercial silica support in Figure 4b, the type H3 hysteresis loop was observed, in contrast to type H2(a) realized in the cornhusk support (Figure 4a). According to Johansson et al. [49], an H3 hysteresis loop is indicative of slit-shaped mesopores or plate-like particles and is mainly observed in materials comprising aggregates (loose assemblage), as can be observed in the SEM micrograph of the Pd/CeO_2_/commercial catalyst in Figure 2d. Conversely, the H2 hysteresis formed in the cornhusk support is due to delayed condensation during adsorption, exhibited by various adsorbents such as inorganic oxide gels and porous vycor glasses, as explained in our earlier study on mesoporous silica materials derived from sol-gel processes [43]. Eventually, the pore volume measured for the Pd/CeO_2_/commercial catalyst appeared higher than that of the Pd/CeO_2_/CHSiO_2_ owing to the loose assemblage of aggregates in the former, as shown in Table 2.

Table 2 summarizes the Brunauer–Emmett–Teller (BET) surface areas and NDLFT pore volumes measured for the unsupported and as-synthesized catalyst samples. Unlike the commercial support, the cornhusk support (silica xerogel) exhibited a bimodal pore system consisting of micropores and mesopores with widths centered at 1.5 and 3.8 nm, respectively, as observed in Appendix A. Further analysis by t-plot showed that the silica xerogel exhibited a micropore volume V_micro_ = 0.12 cm^3^/g and micropore surface area S_micro_ = 211 m^2^/g. This allowed for an optimal distribution of the active catalytic species and improved reaction kinetics. The mesopores within the cornhusk silica support can enhance mass transport and accessibility to reaction sites, while the micropores can retain and stabilize catalytic species [58]. On the other hand, the t-plot kernel showed a non-existence micropore volume and surface area (V_micro_ and S_micro_ = 0) within the pore structure of the commercial silica support, although a small indication of closed pores or inaccessible micropores was found. Thus, conclusions about the existence of only mesopores can be made on the commercial silica support with the mean pore width centered at 5.4 nm in Appendix A.

After the catalyst synthesis, the N_2_-desorption capacity observed in the resultant catalysts showed a decrease in the respective surface areas and pore volumes, as seen in Table 2. This signifies the successful filling of the metal oxide particles into the supports. The highest surface area of 77 m^2^/g was achieved in the final as-synthesized Pd/CeO_2_/CHSiO_2_ catalyst compared with the Pd/CeO_2_/commercial catalyst (56 m^2^/g). The respective reduction in the textural properties of the catalysts was mainly due to the CeO_2_ impregnation into the interspaces (mesopores and micropores), as the effects of Pd loading on the supports can be neglected based on its negligible amount compared with the amount of CeO_2_, as evidenced in the XRD profiles in Figure 5. 

X-ray diffraction (XRD) analysis was carried out to investigate the crystalline structures of the prepared Pd/CeO_2_/CHSiO_2_ catalyst and Pd/CeO_2_/commercial support, as shown in Figure 5.

The XRD patterns of both catalysts in Figure 5 displayed a series of well-defined peaks with the reflection angles at 2θ = 28.6, 33.2, 47.6, 56.4, 59.3, 69.6, 77.1, and 79.3°, attributed to the reflection of the hexagonal phase of CeO_2_ (111), CeO_2_ (200), CeO_2_ (220), CeO_2_ (311), CeO_2_ (222), CeO_2_ (400), and CeO_2_ (311) lattice planes [Joint Committee on Powder Diffraction Standards (JCPDS) card number no. 34-0394], respectively. This indicates that the CeO_2_ nanoparticles are well-dispersed in both catalysts without significant aggregation. However, no peaks related to the PdO phase were observed due to its low content in the catalyst. XRD has detection limits for small percentages of nanoparticles [59]. The broad hump-shaped diffraction peaks between 2θ = 20 and 25° are the characteristics of amorphous silica [60]. The average crystallite sizes of the CeO_2_ phase in the two catalysts were also estimated from the Scherrer equation [48], and the results are summarized in Table 2. The Pd/CeO_2_/CHSiO_2_ catalyst showed a smaller crystallite size (7.2 nm) compared with that of the Pd/CeO_2_/commercial silica support (7.7 nm). 

The temperature-programmed reduction with hydrogen (H_2_-TPR) profiles provide information about the reducibility of the catalysts, which is an important factor in their catalytic activity. The TPR plots of the catalysts were obtained by measuring the H_2_ consumption as a function of temperature, as illustrated in Figure 6. The intense peaks of Pd/CeO_2_/commercial and Pd/CeO_2_/CHSiO_2_ at the lowest temperature of ~30 °C are attributed to the reduction of the PdO_x_ species in contact with CeO_2_ [61]. The TPR plot of the Pd/CeO_2_/CHSiO_2_ catalyst showed two minor peaks, between 200 °C and 400 °C, which corresponded to Pd–O reducibility to Pd and surface reduction of CeO_2_ to CeO_2-x_, respectively [38]. These peaks were rather absent in the Pd/CeO_2_/commercial catalyst, indicating that the Pd–O and CeO_2_ are not well-dispersed on commercial support compared with the cornhusk silica catalyst. In addition, there were broad peaks between 600 and 800 °C, which corresponded to the reduction of bulk CeO_2_ to CeO_2−x_ [62]. 

The differences in reducibility between the two catalysts can be attributed to the different properties of the support materials. The cornhusk support used in synthesizing the Pd/CeO_2_/CHSiO_2_ catalyst had a slightly higher surface area with micropores than that of the commercial silica catalyst (Table 2), which allowed for more Pd/CeO_2_ interaction and a higher metal oxide dispersion on the support, leading to higher catalyst reducibility. Furthermore, by comparing the total amount of H_2_ consumption during the duration of the measurements, Pd/CeO_2_/CHSiO_2_ showed 89% of H_2_ consumption, which was higher than that of the Pd/CeO_2_/commercial (74.1%), denoting higher reducibility compared with the Pd/CeO_2_/commercial [63].

### 3.2. Methane Catalytic Combustion Tests in Dry and Wet Conditions over As-Synthesized Samples: Effect of Support on the Catalytic Activity of Pd/CeO_2_ for Methane Oxidation 

The catalytic performance of the Pd/CeO_2_/CHSiO_2_ and Pd/CeO_2_/commercial catalysts for methane combustion in the temperature range of 150–600 °C was investigated using light-off curves under dry and wet conditions, as shown in Figure 7a,b, respectively. The temperature corresponding to 50% conversion of CH_4_ (T_50%_) is widely used to compare the low-temperature combustion activity of the catalysts [64,65].

The results showed that the Pd/CeO_2_/CHSiO_2_ catalyst exhibited higher catalytic activity in dry conditions than the Pd/CeO_2_/commercial silica catalyst, with a T_50%_ value of 394 °C compared with 593 °C for the commercial catalyst (Figure 7a). The activity of the Pd/CeO_2_/CHSiO_2_ catalyst is comparable to highly efficient catalysts reported in previous studies [66,67,68]. Similarly, Chen et al. [69] reported a catalytic activity for methane combustion with a conversion of 50% at 395 °C for 1 wt.% Pd/Ce supported on Al_2_O_3_.

In addition, the higher catalytic activity exhibited by the Pd/CeO_2_/CHSiO_2_ catalyst may be ascribed to the unique pore structure and surface chemistry, which may have provided more active sites for the catalytic reaction. Likewise, the lower reduction temperature and higher reduction peak area of the Pd/CeO_2_/CHSiO_2_ catalyst, as explained earlier in Figure 6, showed that the Pd/CeO_2_ nanoparticles were more reducible on the cornhusk support, which may have contributed to its enhanced catalytic activity.

The effect of water vapor on the catalytic activity and stability of the prepared catalysts was also investigated in the LTCMC reaction. The CH_4_ conversion over the Pd/CeO_2_/CHSiO_2_ and Pd/CeO_2_/commercial catalysts were performed under 12 vol.% vapor concentrations at a gas hourly space velocity (GHSV) of 87,000 mLg^−1^ h^−1^, and the results are shown in Figure 7b. As observed, the presence of water vapor in the reaction mixture resulted in a significant decrease in the catalytic activities of all the prepared catalysts. Thus, the methane conversion over the catalysts in water vapor decreased from 90 to 73% and 69 to 49% at 600 °C for the Pd/CeO_2_/CHSiO_2_ and Pd/CeO_2_/commercial catalysts, respectively. Similarly, the ignition temperatures of the Pd/CeO_2_/commercial catalyst shifted to higher reaction temperatures, from 315 to 415 °C, in the presence of water vapor. In contrast, the Pd/CeO_2_/CHSiO_2_ catalyst maintained its ignition temperature, albeit with reduced activity. 

The reduced catalytic activity for both as-synthesized catalysts can be attributed to the possibility of the water vapor competing with the methane for active sites on the catalyst surface. This competitive adsorption between the methane and water species leads to blocking the active sites by occupying the surface hydroxyl groups, reducing the catalytic activity [32]. From kinetics studies, the reaction rate for methane oxidation is of order −1 with regard to water concentration, as water prevents the desorption of water from the catalyst surface [70,71].

In addition, water vapor can also lead to the sintering of the metal particles and a decrease in the surface area of the catalyst, which can further reduce the catalytic activity [33]. The catalyst prepared from cornhusk support showed better resistance to water vapor than the commercial catalyst. For example, the methane conversion over the prepared catalyst was still above 70%, while the commercial catalyst showed a methane conversion of approximately below 50% in wet conditions at 600 °C. This observation may be due to the hydrophobic nature of the sol-gel support [42], the high dispersion of the active metal particles, and the strong metal-support interactions in the Pd/CeO_2_/CHSiO_2_ catalyst, which helped to prevent the sintering of the metal particles and maintained the stability of the catalyst under the water vapor atmosphere. Similar adverse effects of water inhibition on catalytic performance have been reported in the literature [32,33]. 

### 3.3. Kinetic Studies 

Kinetic studies were performed on both Pd/CeO_2_/CHSiO_2_ and Pd/CeO_2_/commercial silica catalysts to investigate the reaction mechanism of methane combustion in the simulated synthetic gas flue gas mixture. The activation energies were calculated from the Arrhenius plots, in Equation (2), and estimated within methane conversion values ≤10% to exclude the effects of mass and heat transfer limitations [70,72]. In addition, the Arrhenius plots in Figure 8 were reasonably linear and, thus, allowed reasonable estimations of the activation energies and pre-exponential terms using the Arrhenius Equation (2), which shows the dependence of the rate constant (*k*) on the temperature (*T*) [70].
(2)$$k=K\exp\left[-E_{A}/RT\right]$$

In Equation (2), *K* is the pre-exponential factor, *E_A_* is the activation energy, and *R* is the gas constant. According to the results in Figure 8, both catalysts followed pseudo first-order kinetics, with the Pd/CeO_2_/CHSiO_2_ catalyst showing a lower activation energy (*E_A_* = 130 kJ/mol) than the Pd/CeO_2_/commercial catalyst (*E_A_* = 137 kJ/mol). The lower activation energy of the Pd/CeO_2_/CHSiO_2_ catalyst indicates a high possibility of activation of the reaction at a lower temperature, which is beneficial for energy-saving and cost-effective processes [73], and is consistent with the higher catalytic activity for the Pd/CeO_2_/CHSiO_2_. This lower activation energy of the Pd/CeO_2_/CHSiO_2_ catalyst could be ascribed to the presence of surface hydroxyl groups, which facilitated the activation of methane molecules [74]. 

According to the literature [74], the presence of surface hydroxyl groups on transition metal oxides (TMOx) facilitates methane activation by promoting the dissociation of the C–H bonds. The cornhusk support provides a large surface area and abundant surface hydroxyl groups, as evidenced in the FTIR diagram in Figure 3, enhancing the dispersion of Pd particles and promoting active PdO_x_ species formation. The −OH groups can then act as proton acceptors or donors, creating a favorable environment for the methane reaction. In addition, they can form coordinatively unsaturated metal sites that act as active sites for methane activation. Furthermore, the hydroxyl groups can also promote methane adsorption onto the TMOx surface, increasing methane concentration at the reaction sites, and thus enhancing methane activation [74,75,76,77]. 

In chemical kinetics, the pre-exponential factor, also known as the pre-exponential constant or A factor, is a parameter in the Arrhenius equation representing the frequency of successful collisions between two reactant molecules to form the activated complex [9]. The pre-exponential factor is an important parameter to consider when evaluating the catalytic activity of a material. It is related to the rate constant, indicating the rate at which the reaction proceeds [78]. The pre-exponential factor is influenced by various factors, including the nature and concentration of the reactants, the temperature, and the catalyst used in the reaction [79]. In the case of the Pd/CeO_2_/CHSiO_2_ and Pd/CeO_2_/commercial catalysts, the pre-exponential factor can provide insight into their catalytic activity and efficiency.

The results of the studies have shown that the pre-exponential factor of the Pd/CeO_2_/CHSiO_2_ catalyst is higher (A = 3.2 × 10^13^ h^−1^) than that of the Pd/CeO_2_/commercial catalyst (A = 2.8 × 10^13^ h^−1^), suggesting that the latter had a higher frequency of successful collision between the CH_4_ and O_2_ molecules. This indicates that the Pd/CeO_2_/CHSiO_2_ catalyst is more efficient at promoting the reaction between methane and oxygen and requires lower activation energy to initiate the reaction. This is consistent with the higher catalytic activity observed for the Pd/CeO_2_/ cornhusk catalyst, rendering it a more promising candidate for catalyzing the combustion of methane.

### 3.4. Long-Term Catalytic Tests

Long-term catalytic tests are important to assess the durability and reliability of catalysts in real-world applications. This study evaluated the Pd/CeO_2_/CHSiO_2_ and Pd/CeO_2_/commercial catalysts for their long-term catalytic performance during continuous methane combustion for 10 h under reaction conditions of 500 °C and a space velocity of 87,000 mLg^−1^ h^−1^.

According to the results illustrated in Figure 9, both catalysts exhibited comparable and excellent stabilities. However, the catalytic activity of the Pd/CeO_2_/CHSiO_2_ and Pd/CeO_2_/commercial catalysts degraded gradually with time on stream. For the Pd/CeO_2_/CHSiO_2_ catalyst, the fresh catalyst showed a methane conversion of 68%, which decreased to 61% after the reaction at 500 °C for 10 h, accounting for a 7% reduction in catalytic activity. Comparatively, there was a decrease in catalytic activity for the Pd/CeO_2_/commercial catalyst from 39% to 28%, resulting in an 11% decrease in activity at the same set of experimental conditions. Thus, the Pd/CeO_2_/CHSiO_2_ catalyst exhibited superior long-term stability compared with the Pd/CeO_2_/commercial catalyst, which may be ascribed to the unique properties of the cornhusk support. The high surface area and surface hydroxyl groups of the cornhusk support may have enhanced the metal-support interaction and stability compared with the commercial silica support, which lacked some of these properties. Hence, it is probable that there was a state transformation of active Pd species and a sintering of Pd particles in the Pd/CeO_2_/commercial catalyst during the long-term catalytic tests. Thus, it was tentatively more prone to aggregation and deactivation [80]. For example, the accumulation of reaction intermediates or coke deposition on the catalyst surface could lead to a decrease in the active surface area, resulting in reduced catalytic activity over time. Another possibility could be the deactivation of the catalyst due to changes in the oxidation state or morphology of the palladium nanoparticles during the long-term catalytic reaction. Further characterization techniques, such as in situ spectroscopic or microscopic analysis, could provide insights into the changes occurring at the catalyst surface during the reaction and help elucidate the reasons for the observed phenomenon. Similar results of Pd sintering and deactivation were reported in the study of Yang et al. [80]. 

## 4. Conclusions

In conclusion, this study investigated the performance and characterization of sol-gel-derived cornhusk silica support for low-temperature catalytic methane combustion (LTCMC). The results showed that the cornhusk-derived support exhibited promising properties as a support material for the Pd/CeO_2_ catalyst in LTCMC, with comparable or even better performance than the commercial support. The prepared catalysts were characterized by various techniques, including SEM/EDX, XRD, BET, and H_2_-TPR, which showed the successful synthesis of highly dispersed Pd and CeO_2_ nanoparticles on the cornhusk support. The observed superior performance of the cornhusk support was attributed to its unique properties, including high surface area, high porosity, and stability. The lower activation energy of the prepared catalysts, as indicated by the Arrhenius plot, suggests that the Pd/CeO_2_ catalyst on the cornhusk support facilitated the reaction mechanism of methane combustion. Moreover, the Pd/CeO_2_/CHSiO_2_ catalyst showed good long-term catalytic stability. In a gas stream containing water vapor, the Pd/CeO_2_/CHSiO_2_ catalyst also exhibited better performance than the Pd/CeO_2_/commercial silica catalyst. Overall, this study provides valuable insights into the potential of cornhusk-derived support materials for LTCMC applications. The utilization of agricultural waste materials as support materials could offer a sustainable and cost-effective approach to catalyst synthesis while also contributing to a reduction in waste residues. Further studies could investigate optimizing the cornhusk support properties and exploring other metal oxide catalysts for LTCMC on biogenic supports.

## Figures and Tables

**Figure 1 nanomaterials-13-01450-f001:**
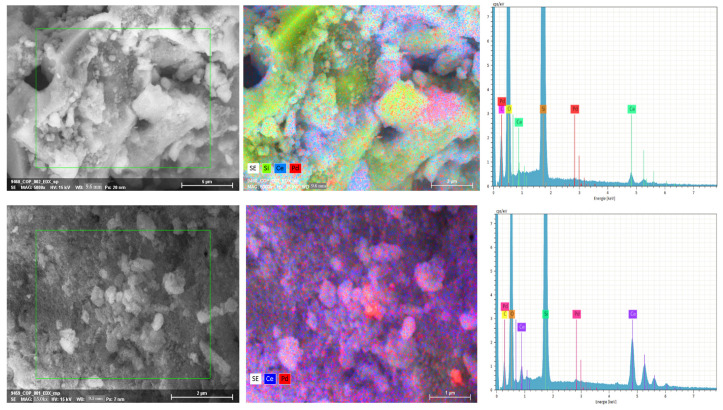
SEM/EDX mapping and spatial distributions of the Pd and Ce nanoparticles on supports. (**Top row**): cornhusk support; (**Bottom row**): commercial silica supports.

**Figure 2 nanomaterials-13-01450-f002:**
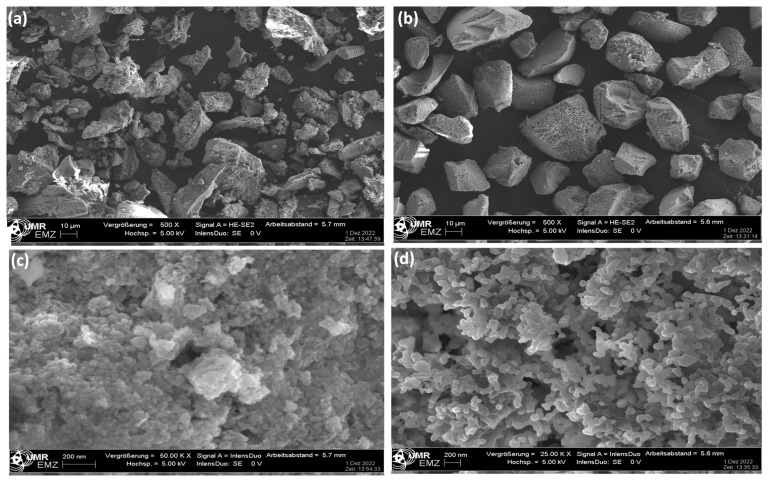
SEM micrographs of synthesized catalysts prepared from cornhusk (**a**,**c**) and commercial (**b**,**d**) silica supports.

**Figure 3 nanomaterials-13-01450-f003:**
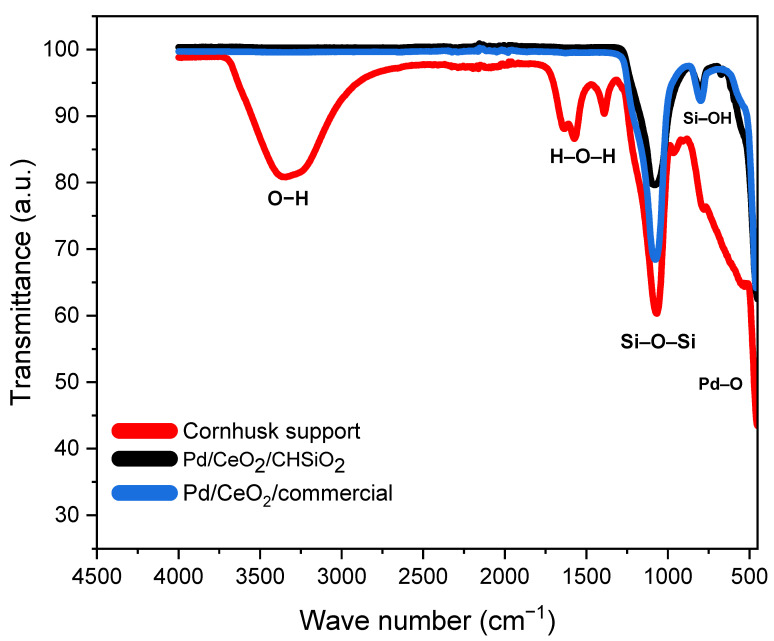
FTIR spectra of cornhusk support material, Pd/CeO_2_/CHSiO_2_, and Pd/CeO_2_/commercial catalysts.

**Figure 4 nanomaterials-13-01450-f004:**
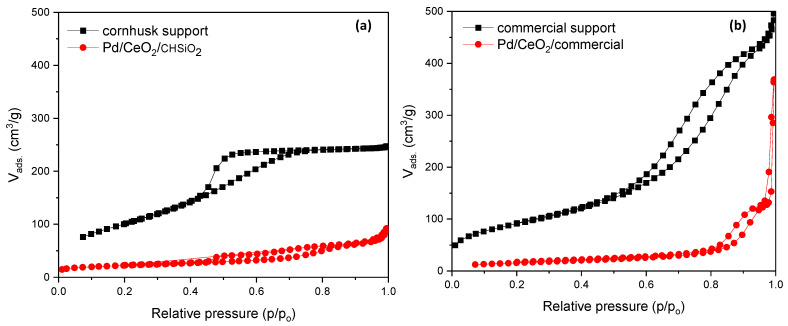
N_2_ adsorption and desorption isotherms of (**a**) cornhusk support and Pd/CeO_2_/CHSiO_2_; (**b**) commercial support and Pd/CeO_2_/commercial.

**Figure 5 nanomaterials-13-01450-f005:**
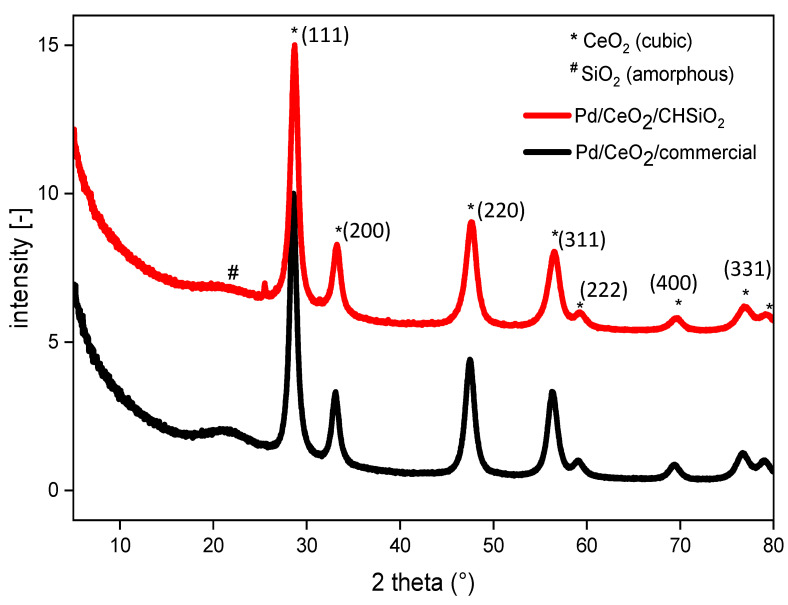
X-ray diffraction pattern for the synthesized Pd/CeO_2_/CHSiO_2_ and Pd/CeO_2_/commercial catalysts.

**Figure 6 nanomaterials-13-01450-f006:**
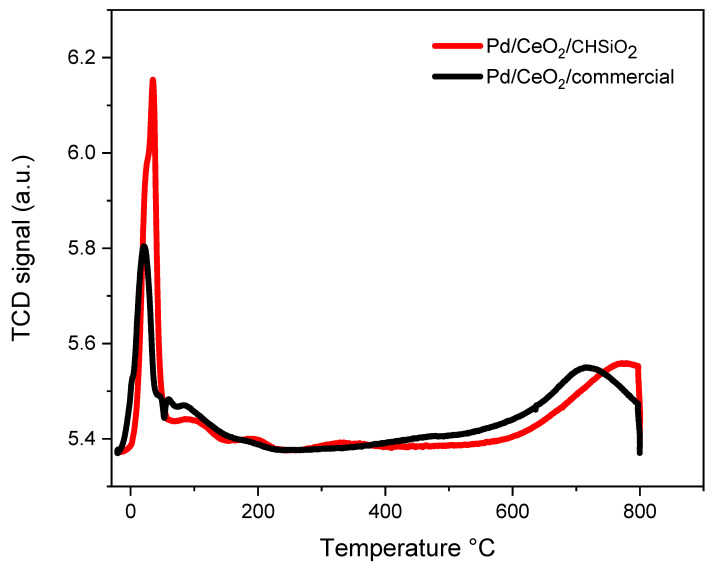
H_2_-TPR profiles of Pd/CeO_2_/CHSiO_2_ and Pd/CeO_2_/commercial catalysts.

**Figure 7 nanomaterials-13-01450-f007:**
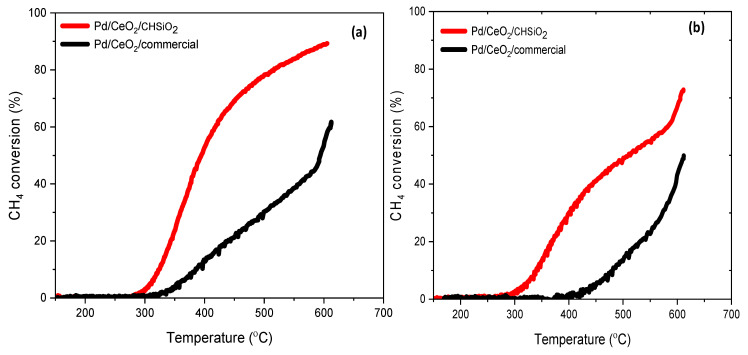
Light-off curves for CH_4_ combustion in stoichiometric conditions; (**a**) dry conditions (**b**) wet conditions. Conditions: Gas mixture containing 800–1000 ppm CH_4_, 1528 ppm CO, 207 ppm NO, 10 vol.%, CO_2_, 6 vol.% O_2_ balanced with N_2_, or a mixture of above compositions + 12 vol.% H_2_O (wet condition) (catalyst mass: 0.2 g mixed with 1.2 g of corundum, a space velocity of 87,000 mLg^−1^ h^−1^ in simulated synthetic gas with total flow rate of 70 mL/min).

**Figure 8 nanomaterials-13-01450-f008:**
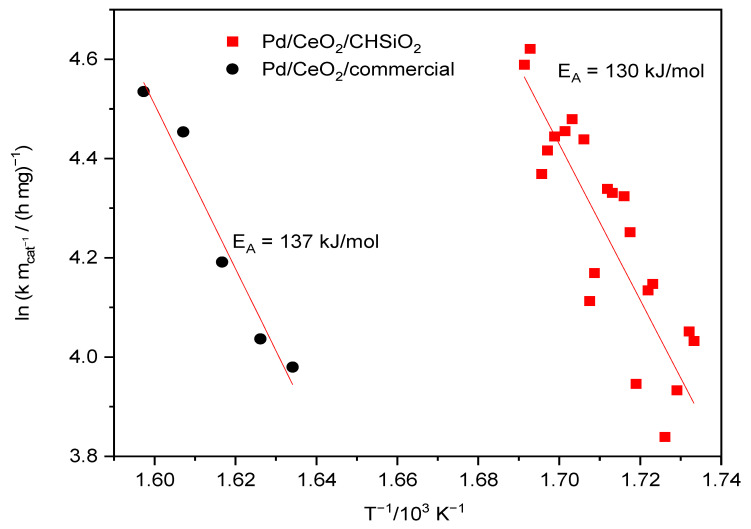
Arrhenius plots over Pd/CeO_2_/CHSiO_2_ and Pd/CeO_2_/commercial catalysts at 600 °C. Conditions: Gas mixture containing 800–1000 ppm CH_4_, 1528 ppm CO, 207 ppm NO, 10 vol.% CO_2_, 6 vol.% O_2_ balanced with N_2_ (dry condition, catalyst mass: 0.2 g mixed with 1.2 g of corundum, a space velocity of 87,000 mLg^−1^ h^−1^ in simulated synthetic gas with total flow rate of 70 mL/min).

**Figure 9 nanomaterials-13-01450-f009:**
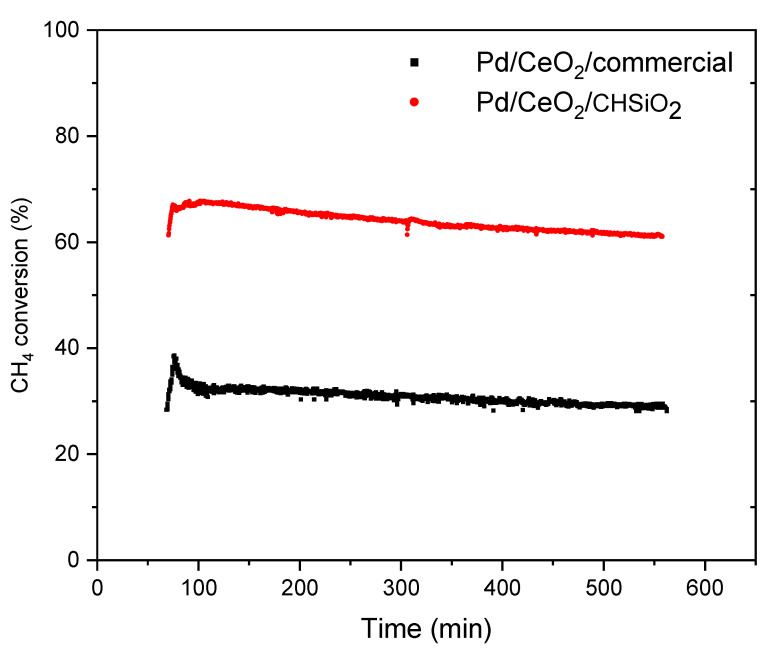
Methane conversion versus reaction time of the catalysts after reaction for 10 h. The reactions were conducted at 500 °C with gas mixture containing 800–1000 ppm CH_4_, 1528 ppm CO, 207 ppm NO, 10 vol.% CO_2_, 6 vol.% O_2_ balanced with N_2_ (dry condition, catalyst mass: 0.2 g mixed with 1.2 g of corundum, a space velocity of 87,000 mLg^−1^ h^−1^ in simulated synthetic gas with total flow rate of 70 mL/min).

**Table 1 nanomaterials-13-01450-t001:** Elemental compositions of catalyst samples measured by ICP-OES analysis (oxygen neglected).

Catalyst	Elements ^a^		
	Si %	Ce %	Pd %
Pd/CeO_2_/CHSiO_2_	20.1	29.9	0.92
Pd/CeO_2_/commercial	20.9	32.4	1.0

^a^ Calculated bulk atomic composition.

**Table 2 nanomaterials-13-01450-t002:** Textural properties of the silica support and synthesized catalysts. Specific surface area (S_BET_), pore volume (NLDFT, determined by N_2_-sorption), micropore volume (V_micro_), micropore surface area (S_micro_, determined by t-plot method), and external surface area (S_ext_, determined by t-plot method).

Sample	S_BET_ (m^2^/g)	V_t_ (cm^3^/g)	V_micro_ (cm^3^/g)	S_micro_ (m^2^/g)	Sext (m^2^/g)	Ce ^a^ (nm)
Cornhusk support	384	0.35	0.12	211	173	-
Commercial support	329	0.66	-	-	-	-
Pd/CeO_2_/CHSiO_2_	77	0.11	0.01	12	65	7.2
Pd/CeO_2_/commercial	56	0.2	-	-	-	7.7

^a^ The average crystallite sizes of the CeO_2_ phase in the two catalysts estimated from the Debye–Scherrer equation [48] and the representative reflections in the XRD diagram (Figure 5) at full-width half maximum.

## Data Availability

The data are included in the main text and in the Appendix A.

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
