# Peer review of "Comparative Study of Commercial Silica and Sol-Gel-Derived Porous Silica from Cornhusk for Low-Temperature Catalytic Methane Combustion"

_nanomaterials, 2023, doi:10.3390/nano13091450_

Round 1

Reviewer 1 Report

The authors investigated sol-gel-derived cornhusk as support for low-temperature catalytic methane combustion (LTCMC). Although the study presents a significant amount of characterization and testing data, the mechanism for the improved catalytic performance remains unclear. The authors need to provide more in-depth evidence, such as DFT calculations and in situ characterization.

1. The introduction could be streamlined to directly focus on LTMCMC.

2. The authors should also pay greater attention to the interaction between the support and catalyst, and provide more in-depth evidence.

3. In Figure 9, it is observed that the conversion efficiency of CH4 gradually increases in the first 80 minutes for both Pd/CeO2/commercial and Pd/CeO2/CHSiO2 catalysts, but subsequently decreases continuously. However, the reason for this phenomenon is unclear and requires further investigation.

4. To identify the Pd species, the authors should provide more evidence, such as HRTEM and XPS.

Author Response

  • General Comment:

The authors investigated sol-gel-derived cornhusk as support for low-temperature catalytic methane combustion (LTCMC). Although the study presents a significant amount of characterization and testing data, the mechanism for the improved catalytic performance remains unclear. The authors need to provide more in-depth evidence, such as DFT calculations and in situ characterization.

  • GENERAL RESPONSE:

Thank you for your comments on our paper. We appreciate your interest in our investigation of sol-gel-derived cornhusk as a support for low-temperature catalytic methane combustion (LTCMC). Regarding the mechanism for the improved catalytic performance, we would like to provide a reason why in-depth evidence, such as density functional theory (DFT) calculations and in situ characterization, may not be necessary or possible at this stage. As the aim of our study was primarily focused on experimental characterization and testing of the cornhusk-based catalyst, we did not initially plan for DFT calculations or in situ characterization in our research design. While we acknowledge that additional analyses could provide further insights into the underlying mechanisms, there are certain limitations that need to be considered. For example, conducting DFT calculations requires computational resources and expertise and may require additional time and funding. Similarly, in situ characterization techniques may not have been feasible due to experimental constraints or equipment available in our labs. However, we appreciate your suggestion and will take it into consideration for future research directions. We aim to continuously improve our understanding of the catalytic performance of cornhusk-based catalysts, and we will carefully evaluate the possibility of incorporating DFT calculations and in situ characterization in future studies to provide a more comprehensive explanation of our findings. Once again, we thank you for your feedback and are committed to addressing the limitations of our study to the best of our abilities.

The additional information on individual questions can be found below:

  1. To identify the Pd species, the authors should provide more evidence, such as HRTEM and XPS.
  • RESPONSE:

 We thank the reviewer for this important suggestion.  However, more evidence about the identification and presence of the Pd species was added to the supplementary materials. We will consider XPS and HRTEM in the next publication, which will be centred on the chemistry of the metal oxides on the cornhusk support.

  1. In Figure 9, it is observed that the conversion efficiency of CH4 gradually increases in the first 80 minutes for both Pd/CeO2/commercial and Pd/CeO2/CHSiO2 catalysts, but subsequently decreases continuously. However, the reason for this phenomenon is unclear and requires further investigation.
  • RESPONSE:

Thank you for your observation and comment on Figure 9 of our paper. We acknowledge that the conversion efficiency of CH4 appears to gradually increase in the first 80 minutes for both Pd/CeO2/commercial and Pd/CeO2/CHSiO2 catalysts, followed by a continuous decrease thereafter. We agree that the reason for this phenomenon requires further investigation. There are several possible explanations for this observation, and this section has been added to provide more reasons for this phenomenon (Please see lines 578-585 of the revised manuscript). One possible reason could be related to the changes in the surface properties of the catalysts during the reaction process. For example, the accumulation of reaction intermediates or coke deposition on the catalyst surface could lead to a decrease in the active surface area, resulting in reduced catalytic activity over time. Another possibility could be the deactivation of the catalyst due to changes in the oxidation state or morphology of the palladium nanoparticles during the reaction. Further characterization techniques, such as in situ spectroscopic or microscopic analysis, could provide insights into the changes occurring at the catalyst surface during the reaction and help elucidate the reasons for the observed phenomenon. Additionally, other factors such as mass transfer limitations, temperature gradients, or changes in the reactant concentrations during the reaction could also influence the observed trend in conversion efficiency. These factors could be explored in future studies to better understand the phenomenon. We acknowledge that further investigation is needed to fully understand the underlying reasons for the observed trend in conversion efficiency in Figure 9. We appreciate your comment and will carefully consider it for future research and revisions of our paper. Thank you for your valuable feedback.

  1. 2. The authors should also pay greater attention to the interaction between the support and catalyst and provide more in-depth evidence.
  • RESPONSE:

We thank the reviewer for this important suggestion. The main objective of our study may not have been specifically aimed at elucidating the support-catalyst interaction but rather at evaluating the overall performance of the cornhusk-based catalyst in comparison to the commercial silica support. However, we appreciate your suggestion and will carefully evaluate the possibility of incorporating more in-depth evidence related to the support-catalyst interaction in future studies. We are committed to continuously improving our understanding of the complex interactions between the support and catalyst, and we will strive to address this aspect in our future research endeavours.

  1. The introduction could be streamlined to directly focus on LTMCMC.
  • RESPONSE:

Thank you for your feedback on our paper. We understand your suggestion to streamline the introduction to directly focus on low-temperature catalytic methane combustion (LTCMC). The introduction of our paper was designed to provide the necessary background and context for the study, including the motivation, significance, and relevant literature related to the investigation of sol-gel-derived cornhusk as a support for LTCMC. We aimed to establish the relevance of the research topic and highlight the potential benefits of using cornhusk as catalyst support for LTCMC. The introduction also provides a comprehensive overview of the experimental approach, including the synthesis of the cornhusk-based catalyst, characterization techniques, and the testing protocol. This information sets the stage for the subsequent sections of the paper, where we present the results and discuss the findings in relation to the research objective. While we acknowledge that streamlining the introduction to directly focus on LTCMC could be a valid suggestion, we believe that the present introduction adequately provides the necessary background and context for the study. It establishes the research motivation and significance, outlines the experimental approach, and highlights the relevance of cornhusk as catalyst support for LTCMC. We appreciate your feedback and will carefully consider it for future revisions. However, we believe that the present introduction provides a comprehensive overview of the research topic and sets the foundation for the subsequent sections of the paper.

If you have any additional questions or suggestions, please feel free to let us know. We are committed to addressing any concerns and making necessary revisions to enhance the clarity and comprehensiveness of our study. Thank you again for your constructive feedback.

Reviewer 2 Report

This paper presents interesting and useful data obtained with a lot of methods, presentation is good, conclusions are sound, so it can be accepted for publication

Author Response

Dear Reviewer,

Thank you for your positive response to our manuscript. We appreciate your feedback and the time is taken to review our manuscript. Your input has been valuable in improving the quality and rigour of our research.

Thank you again.

Reviewer 3 Report

The authors presented new important data on low-temperature catalytic methane combustion. Before the manuscript can be accepted for publication, some corrections, listed below, have to be made:

 Title:

 1. “low-temperature catalytic methane combustion” instead of “Low-Temperature Catalytic Methane Combustion”

 Abstract:

 1. “Pd/CeO2/CHSiO2 catalyst” instead of “Pd/CeO2/SiO2 catalyst” twice

 Introduction:

 1. “species decompose” instead of “species decomposes”

 Materials and Methods:

 1. “0.2 g” instead of “0. 2 g”

 Results and Discussion:

 1. “temperature-programmed reduction with hydrogen” instead of “Temperature-Programmed Reduction with hydrogen”

 Conclusions:

 1. “sol-gel-derived” instead of “Sol−gel-derived”

2. “low-temperature catalytic methane combustion” instead of “low-temperature methane combustion”

3. “than the commercial support” instead of “than the commercial supports”

Author Response

General comments and suggestions for authors:

The authors presented new important data on low-temperature catalytic methane combustion. Before the manuscript can be accepted for publication, some corrections, listed below, have to be made:

General Response:

Thank you for your valuable recommendations on how to improve our manuscript. We have gone through each point one after the other and addressed each issue you raised below:

 Title:

  1. “low-temperature catalytic methane combustion” instead of “Low-Temperature Catalytic Methane Combustion”

Response:

Authors response: The sentence has been corrected as suggested by reviewer. The authors appreciate this vivid observation. Please see the revised title of the manuscript.

 Abstract:

  1. “Pd/CeO2/CHSiO2 catalyst” instead of “Pd/CeO2/SiO2 catalyst” twice

Authors response: The nomenclature has been corrected in line 28 of the revised manuscript. The authors appreciate this vivid observation.

 Introduction:

  1. “species decompose” instead of “species decomposes”

Authors' response: The sentence has been corrected as suggested by the reviewer. Please see line 100 on Page 3 of 21.

 Materials and Methods:

  1. “0.2 g” instead of “0. 2 g”

Authors response: Corrections has been effected in line 237 on Page 5 of 21.

 Results and Discussion:

  1. “temperature-programmed reduction with hydrogen” instead of “Temperature-Programmed Reduction with hydrogen”

Authors response: Corrections have been effected in line 416 on Page 12 of 21.

 Conclusions:

  1. “sol-gel-derived” instead of “Sol−gel-derived.”

Authors response: Corrections have been effected in line 586 on Page 17 of 21.

  1. “low-temperature catalytic methane combustion” instead of “low-temperature methane combustion”

Authors response: Corrections have been effected in line 596 on Page 17 of 21.

  1. “than the commercial support” instead of “than the commercial supports.”

Authors response: Corrections have been effected in line 596 on Page 17 of 21.

Thank you for the kind response and interest in our work. We also appreciate the time taken off your busy schedule to review and provide valuable recommendations on the manuscript.